# The Molecular Effect of Wearing Silver-Threaded Clothing on the Human Skin

Alexey V. Melnik,[a,g,h] Chris Callewaert,[b, i] Kathleen Dorrestein,[a] Rosie Broadhead,[i] Jeremiah J. Minich,[k] Madeleine Ernst,[a,l] Greg Humphrey,[b] Gail Ackermann,[b] Rob Gathercole,[c] Alexander A. Aksenov,[a,g,h] Rob Knight,[b,d,e,j] Pieter C. Dorrestein[a,b,e,f]

aCollaborative Mass Spectrometry Innovation Center, Skaggs School of Pharmacy and Pharmaceutical Sciences, University of California San Diego, La Jolla, California, USA

bDepartment of Pediatrics, University of California San Diego, La Jolla, California, USA

cProduct Innovation team, Lululemon, Vancouver, Canada

dDepartment of Computer Science and Engineering, University of California, San Diego, La Jolla, California, USA

eCenter for Microbiome Innovation, University of California, San Diego, La Jolla, California, USA

fDepartment of Pharmacology, University of California, San Diego, La Jolla, California, USA

gDepartment of Chemistry, University of Connecticut, Storrs, Connecticut, USA

hArome Science Inc., Farmington, Connecticut, USA

iCenter for Microbial Ecology and Technology, Ghent University, Ghent, Belgium

jDepartment of Bioengineering, University of California San Diego, La Jolla, California, USA

kPlant Molecular and Cellular Biology Laboratory, The Salk Institute for Biological Studies, La Jolla, California, USA

lSection for Clinical Mass Spectrometry, Danish Center for Neonatal Screening, Department of Congenital Disorders, Statens Serum Institut, Copenhagen, Denmark

Alexey V. Melnik, Chris Callewaert, and Kathleen Dorrestein are joint first authors. Author order was decided based on contribution to manuscript writing and preparation.

**ABSTRACT**  With growing awareness that what we put in and on our bodies affects our health and wellbeing, little is still known about the impact of textiles on the human skin. Athletic wear often uses silver threading to improve hygiene, but little is known about its effect on the body's largest organ. In this study, we investigated the impact of such clothing on the skin's chemistry and microbiome. Samples were collected from different body sites of a dozen volunteers over the course of 12 weeks. The changes induced by the antibacterial clothing were specific for individuals, but more so defined by gender and body site. Unexpectedly, the microbial biomass on skin increased in the majority of the volunteers when wearing silver-threaded T-shirts. Although the most abundant taxa remained unaffected, silver caused an increase in diversity and richness of low-abundant bacteria and a decrease in chemical diversity. Both effects were mainly observed for women. The hallmark of the induced changes was an increase in the abundance of various monounsaturated fatty acids (MUFAs), especially in the upper back. Several microbe-metabolite associations were uncovered, including *Cutibacterium*, detected in the upper back area, which was correlated with the distribution of MUFAs, and *Anaerococcus* spp. found in the underarms, which were associated with a series of different bile acids. Overall, these findings point to a notable impact of the silver-threaded material on the skin microbiome and chemistry. We observed that relatively subtle changes in the microbiome result in pronounced shifts in molecular composition.

**IMPORTANCE**  The impact of silver-threaded material on human skin chemistry and microbiome is largely unknown. Although the most abundant taxa remained unaffected, silver caused an increase in diversity and richness of low-abundant bacteria and a decrease in chemical diversity. The major change was an increase in the abundance of various monounsaturated fatty acids that were also correlated with *Cutibacterium*. Additionally, *Anaerococcus* spp., found in the underarms, were associated with different bile acids in the armpit samples. Overall, the impact of the silver-threaded clothing was gender and body site specific.

**KEYWORDS**  clothing, outfit, silver-threaded clothing, skin chemistry, skin microbiome

Address correspondence to Rob Knight, rknight@ucsd.edu, or Pieter C. Dorrestein, pdorrestein@ucsd.edu.

The authors declare a conflict of interest. R.G. is a part of the company Lululemon, which provided the material for the study and partially sponsored the research. P.C.D. is a scientific advisor for Sirenas, Cybele, and Galileo and a scientific advisor and founder of Enveda and OMETA with approval by University of California, San Diego. A.A.A. and A.V.M. are co-founders of Arome Science Inc.

In recent years, there has been a fundamental shift in our understanding of the innate chemistry of the body and the role the microbiome plays in shaping it. Individual chemistries, the molecular makeup of our bodies, are influenced by a multitude of genetic and environmental factors. Some are known to have a profound influence (in particular, diet), while the roles of other factors such as the clothing and products we wear on our skin are less well explored (1–3). The skin is home to an estimated $10^{12}$ microorganisms that live and feed on the skin secretions (4). These billions of microorganisms, including bacteria, fungi, protists, and viruses, form the so-called skin microbiome (5, 6). The role of the microbiome, in particular, has been uncovered, time and time again, as a key factor in a number of pathologies (7, 8). A community of microorganisms creates an environment that enables the skin to be less susceptible to diseases and pathogenic invasion, and therefore, microbial diversity and a healthy immune system are inherently linked (3, 6, 7).

There has been an appreciation for the role of skin microbes and their interaction with clothing for aspects such as body malodor (2). Yet, the ways and effects of altering skin microbiota with clothing and the corresponding effects on health and wellbeing are not known but are a growing subject of research (9). In large part due to the activity of the microbiome, the skin contains various low-molecular-weight compounds (metabolites). Some of these compounds are by-products of either endogenous or microbial metabolism, and others are implemented as part of the skin's physiological role (10). The metabolome along with the microbiome represents the body's first barrier to external and environmental substances. Through these metabolic pathways the skin transfers topical signals to determine the body's physiological activities and regulate homeostasis. Thereby, these processes are adapted to various external factors, such as cosmetics, clothing, or other environmental influences. Many of the same enzymes which operate in the liver also inhabit the skin, and as such, the skin is an important metabolically active organ (11, 12). The skin surface can be sampled to detect and quantify skin metabolites related to diseases, through secreted sweat. Through techniques such as chromatography-mass spectrometry (MS) and other methods, broad-spectrum skin metabolite specimens can be characterized (10). These biomarkers are an important tool for the diagnosis and treatment of skin diseases and how environmental factors may be of influence (13, 14).

Consequently, skin metabolome and microbiome analysis can be a useful indicator in the investigation of the effects of various external stimuli. One of these external stimuli is the incorporation of antibacterial agents into a fabric, for example, the use of metallic and synthetic antimicrobial biocides for odor control in textiles. The increase of antimicrobial agents in our everyday products including our clothing has made it difficult to reestablish or maintain the beneficial bacteria that the body would regularly be exposed to in a more natural environment (15). One of the most common antimicrobial additives is the incorporation of silver ions or nanoparticles. Silver has broad-spectrum antibacterial properties against Gram-positive and -negative bacteria. In particular, X-Static textiles, silver-coated-yarn-threaded fabrics, are currently used in sports clothing for odor control, hygiene, and social comfort, which can thereby enhance product performance. Overall, there has been an increased demand for the antibacterial effects of metal ions, such as silver in the textiles industry (16–19).

However, the influence of such antimicrobial clothing on the skin microbiome and, especially, the metabolome is largely unknown. By learning about the resulting perturbations of microbial communities or body chemistries, it may be possible to then manipulate the effects, including those on health, by designing the composition of clothing. In this study, we have analyzed skin metabolome and microbiome throughout several weeks of wearing the silver-coated-yarn-threaded T-shirt to document the changes caused by the antibacterial effect of silver. Several studies have analyzed the microorganisms associated with body odor and the bacteria present on malodorous textiles (20, 21). This is the first study to explore the influence of the antibacterial properties of silver-threaded textiles on the skin microbiome and metabolome and their

subsequent influence on the textile microbiome and metabolome. The skin's unique microenvironments were also observed through multiple sample locations across the body. This is an important area of research to determine how antimicrobial textiles affect both the skin microbiome and chemistry and how these two environments are interlinked.

## RESULTS

**Metabolome results.** This study aimed to investigate the impact of antibacterial silver-threaded clothing material on the human skin metabolome and microbiome in a controlled manner. The experiment took place over the course of 12 weeks for a total of 12 volunteers: six males and six females (see Fig. S1 in the supplemental material). Four body sites, chest, upper back, lower back, and armpits, were chosen for sampling because of the most immediate contact with the clothing material (Fig. S1b). As described in Materials and Methods, the experiment longitudinally spanned a total of four phases: initial washout, silver-threaded T-shirts, regular T-shirts, and silver-threaded T-shirts again (Fig. S1a). During the study course, volunteers were prompted to use only the provided skin products to minimize variability. The full sample set included samples from skin, clothing, and the skin products that volunteers had been using.

We then conducted untargeted liquid chromatography-tandem mass spectrometry (LC-MS/MS) analysis as described in Materials and Methods. The resultant data were subjected to molecular networking on Global Natural Products Social Molecular Networking (GNPS) (22). The molecular networking represents all unique compounds as network nodes, and those with similar MS/MS spectra are connected by edges. Since structurally similar compounds tend to also have similar fragmentation patterns, molecular networking enables visualizing chemical relationships within the data set. Figure 1A and B shows molecular networks of all compounds that were detected in this study. Coloring the nodes according to, for example, T-shirt phase (Fig. 1A) or different body parts (Fig. 1B), gives visualizations of associated molecular distributions.

Our results indicate that silver-threaded fabric indeed induces changes of skin chemistry that are detectable with an untargeted approach used in this study. Several observations could be made: one interesting finding is a notable decrease in the chemical diversity for samples of all volunteers except one during the silver shirt usage phase (Fig. S2c). When combined, the differences in Shannon diversity of metabolites were significant and were more pronounced for females ($P = 1.9e-05$) than for males ($P = 0.0027$) (Fig. S2a and Fig. S9). The amount of X-Static silver-threaded yarn was 5% for the female shirt and 4% for the male shirt, mentioned below in Materials and Methods. A persistent challenge in skin studies is the overwhelming signal contribution from skin products, especially deodorants (23). Since the application of deodorants is mostly limited to armpits, we have further considered the body parts separately. The highest impact on chemical diversity was observed for lower and upper back ($P = 1.1e-04$ and 0.0038, respectively), while no significant effect was found on chest ($P = 0.26$) and armpit ($P = 0.42$) (Fig. S2b, Fig. S9 and S10).

We have visualized the metabolome on a principal-coordinate analysis (PCoA) plot with the Canberra dissimilarity metric using EMPeror software (24) (Fig. S3). The main factor driving differences is the sampled body part, which appears to be even more significant than the sampled subject (Fig. S3a). Interestingly, armpit samples spanned the full PC1 space and were separated into two distinct clusters along the PC2 and PC3 axes interpreted as two armpits (Fig. S3b and c). The textile and the skin chemistries differ in the underarms compared to the textile worn next to it (Fig. S3c). Notably, a clear separation of left and right axillae was observed for 8 out of 12 volunteers (Fig. S3c). This discordance was also confirmed by the diversity analysis—the higher chemical diversity in the left armpits of these eight volunteers was observed for both males and females ($P = 0.023$) (Fig. S9d). It is known that there exists a fraction of the population that has distinct microbiomes in their left and right axillae (25). A number of compound classes that may be linked to compounds of bacterial origin—such as acylcarnitines (organonitrogen compounds),

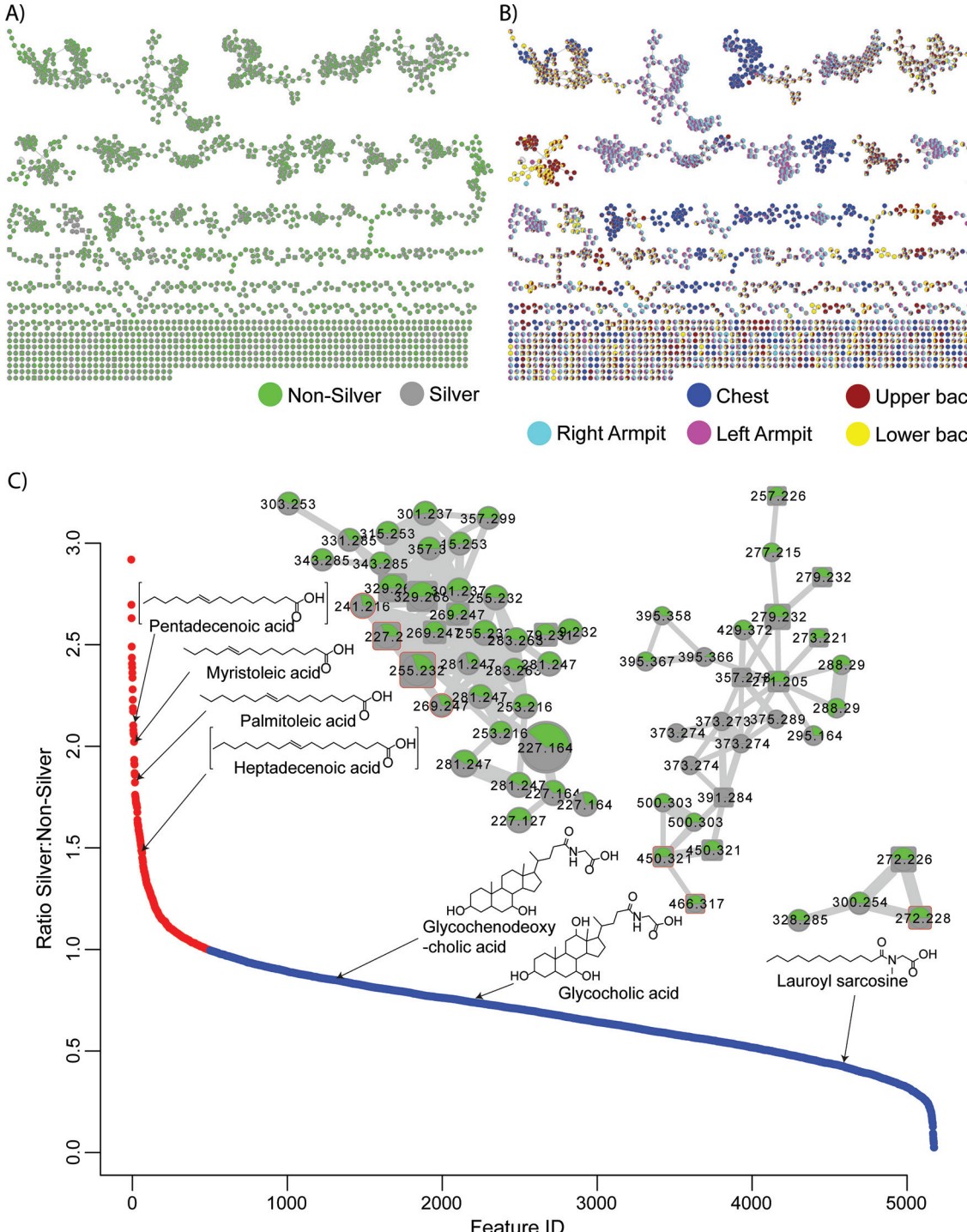

**FIG 1** Molecular networks for exploration of metabolome changes induced by silver fabric. (A and B) Global molecular networks of metabolomics data color coded by T-shirt phase (A) and body part (B). (C) Plot showing the ratio of abundances of metabolites during the silver versus nonsilver phases on upper and lower back. Ratios higher and lower than one (i.e., molecules that increased and decreased in abundance) are highlighted in red and blue, respectively. Examples of some of the annotated molecules with the corresponding clusters from the molecular network that contain them are shown: monounsaturated fatty acids, bile acids, and a surfactant. Clusters are colored the same way as in panel A. Numbers inside cluster nodes denote *m/z* as measured by a mass spectrometer, while the size of the nodes is determined by the peak area under the curve. The thickness of the edges connecting nodes increases continuously with cosine similarity score increase. Consistent ratios of MUFAs' abundances across the cluster are suggestive of the same chemical forces responsible for the changes in their differences in silver versus nonsilver samples. The depicted compounds are highlighted by square nodes.

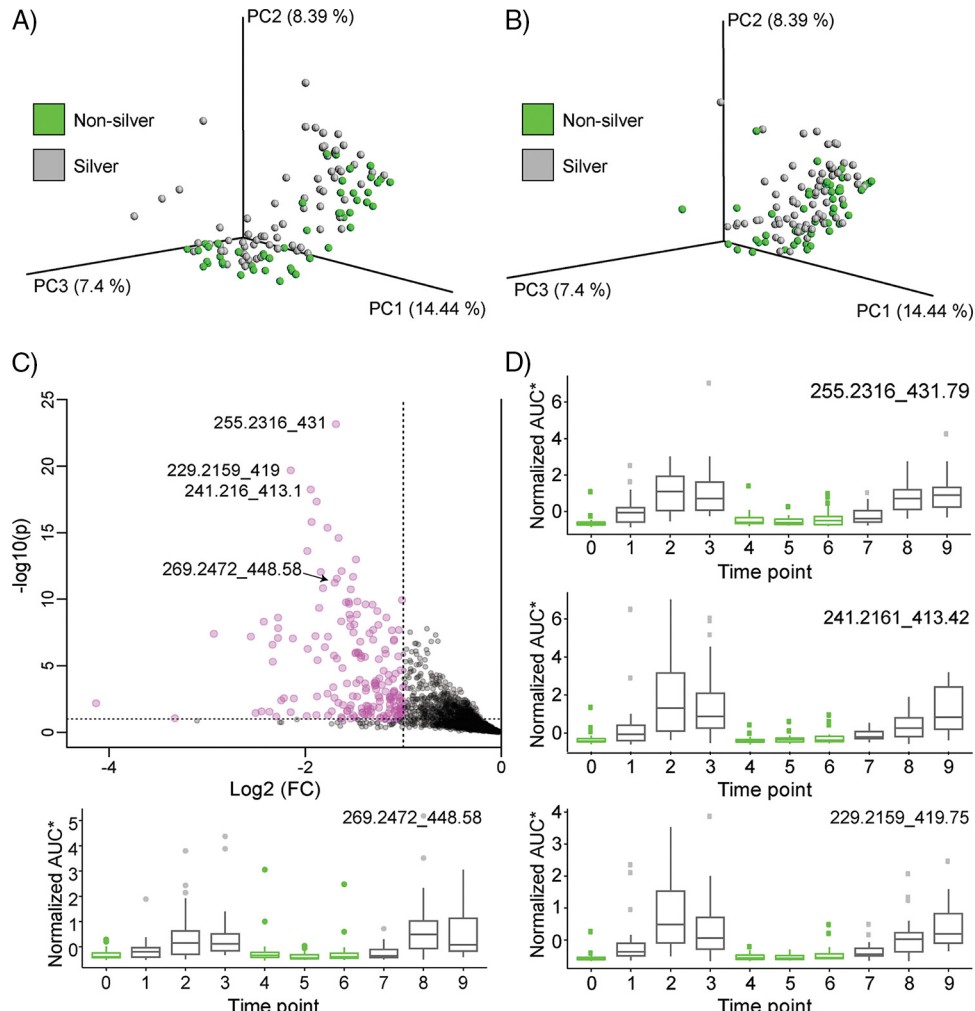

**FIG 2** Monounsaturated fatty acids are the key compounds on skin affected by silver. (A and B) PCoA plots (Canberra distance) of metabolomics data for upper and lower back, respectively. (C) Volcano plot for upper and lower back samples showing the most significant metabolites enriched by silver. The features for some MUFAs are labeled (parent $m/z$_retention time). FC, fold change. (D) Box plots plotted over a time period of sampling for several MUFAs in combined upper and lower back samples. Each box plot title on the top right shows the detected feature with $m/z$_RT, where $m/z$ is the mass as determined by a mass spectrometer and RT is the retention time of the compound on the column during the reverse-phase separation (see Materials and Methods for details). AUC, area under the curve.

glycerophospholipids, and bile acids (steroids and steroid derivatives)—have been found to have different abundances in the two armpits (Fig. S4). Using random forest analysis, we have confirmed that antiperspirant constituents are more abundant in the left armpits, possibly due to more vigorous application by dextral than sinistral volunteers. This finding was corroborated by matched clothing armpit samples (Fig. S3a). Based on our data, we hypothesized that the contribution to the discrepancy in the chemical diversity is due to the chemical constituents of deodorant but also to differences in microbial ecosystems. For both types of participants, with concordant and discordant chemical signatures in axillae, the silver-threaded clothing did not induce differences in axilla chemistries that could be observed with the Canberra distance metric.

When considering body parts separately in unsupervised analysis, the effects of silver cannot be clearly observed: Fig. 2A and B shows PCoA plots for the upper and lower back samples with some separation of samples based on silver and nonsilver groups. Figure 2C shows the volcano plot for these samples. Several features appeared to be significant in discriminating silver and nonsilver sample groups The features were predominantly

attributed to a single network cluster shown in Fig. 1C, indicating their structural similarity. The annotations for these compounds were then established as a variety of monounsaturated fatty acids (MUFAs). The key drivers of the differences include palmitoleic acid, myristoleic acid, and pentadecenoic acid. All of these compounds change in unison in response to wearing silver-threaded T-shirts and were detected in similar abundances (Fig. 2D). The relative abundance of pentadecenoic acid ($C_{15}$ MUFA) increased during the silver T-shirt phase, while its abundance decreased during nonsilver T-shirt phase (Fig. 3D). The presence of odd-carbon short-chain fatty acids (OCS-FAs) among these discriminating compounds strongly suggests their microbial origin. From our data, it appears that silver fabric alters the bacterial populations on skin in a way that results in accumulation of MUFAs.

**Microbiome results.** The paired microbiome analysis has yielded several observations that help to understand the metabolomics findings. Counterintuitively, the skin and textile bacterial communities had a higher bacterial biomass when silver T-shirts were worn than when nonsilver T-shirts were worn ($P = 0.0011$, Mann-Whitney U test) (Fig. 3A and Fig. S5c and d). It was found that the left and right armpits had higher bacterial counts than all other samples and did not differ from each other (Fig. S5a). Yet, the results were different for every individual, with 10 of the 12 volunteers showing more skin bacteria during the silver phase, one volunteer showing no change, and one showing fewer bacteria (Fig. S5b). This is unexpected, considering that the antibacterial properties of silver ions have been widely reported (26). It has been shown that bacteria are readily transferred onto the shirt material in the armpit region (2). Silver textiles can manipulate the microbiome and metabolome of the skin but, importantly, do not reduce microbial biomass. Additionally, it is known that resident microbial strains which cause body odor are due to the presence of certain microbes rather than to biomass (20). The high interindividual response in the skin microbiome was further backed by the initial individual skin microbiome differences (Fig. S9).

Microbe-metabolite cooccurrences were explored using microbe-metabolite vector cooccurrence analysis (mmvec) (Fig. 3), a neural network-based approach that aims to predict metabolite abundances given the presence of a single microbe, and vice versa (27). We detected a range of primary bile acids in the underarms of participants, including glycocholic acid and glycochenodeoxycholic acid. Such bile acids were not detected in other skin body sites. Bile acids were detected in the underarms of all participants, although high interindividual differences were seen in bile acid concentration. The mmvec plot shows the cooccurrence of *Anaerococcus* spp. (Fig. S8a and c) and these bile acids (Fig. 3E). All identified bile acids were significantly correlated with relative abundance of *Anaerococcus* spp. ($P = 4.903e-05$, Spearman correlation) (Fig. 3C and Fig. S8a) and were more abundant in the armpits than on the back and chest (Fig. S8d). *Anaerococcus* spp. were similarly more abundant on armpit skin than on chest skin ($P = 0.00027$) and the back (Fig. S8c). Interestingly, the bile acids were generally decreased in the silver phase (Fig. S8b).

As noted above, during the silver T-shirt phase, we have identified enrichment of a series of monounsaturated fatty acids (MUFAs), particularly on the skin of the chest, upper back, and lower back, as a hallmark feature (Fig. 2), and the presence of odd-number carbon acids indicates that at least some of these compounds can originate only from food or bacterial metabolism. From mmvec analysis, *Corynebacterium* spp. were found to be some of the species that tend to cooccur with MUFAs (Fig. 3F). Other microbial species also appear to play a role in the increase of MUFAs during the silver phase. For example, we found a strong correlation of MUFA abundances (myristoleic acid) with the relative abundance of *Cutibacterium* spp. ($P < 2.2e-16$, Spearman test) (Fig. S6a and f), suggesting these bacteria may also be involved in their biotransformation. Random forest regression analysis showed the highest contribution coming from *Cutibacterium*, compared to all other bacterial taxa (Fig. S6e). We found that *Cutibacterium* spp. were enriched with silver T-shirts both on T-shirts ($P = 0.04$, Mann-Whitney U test) (Fig. 3B and Fig. S6c) and on the skin itself (linear discriminant analysis [LDA] score increase = 2.7) (Fig. S6c). We also found a correlation between $C_{14}$ MUFA ($P = 0.0124$, Spearman test) and $C_{17}$ MUFA ($P = 2e-04$,

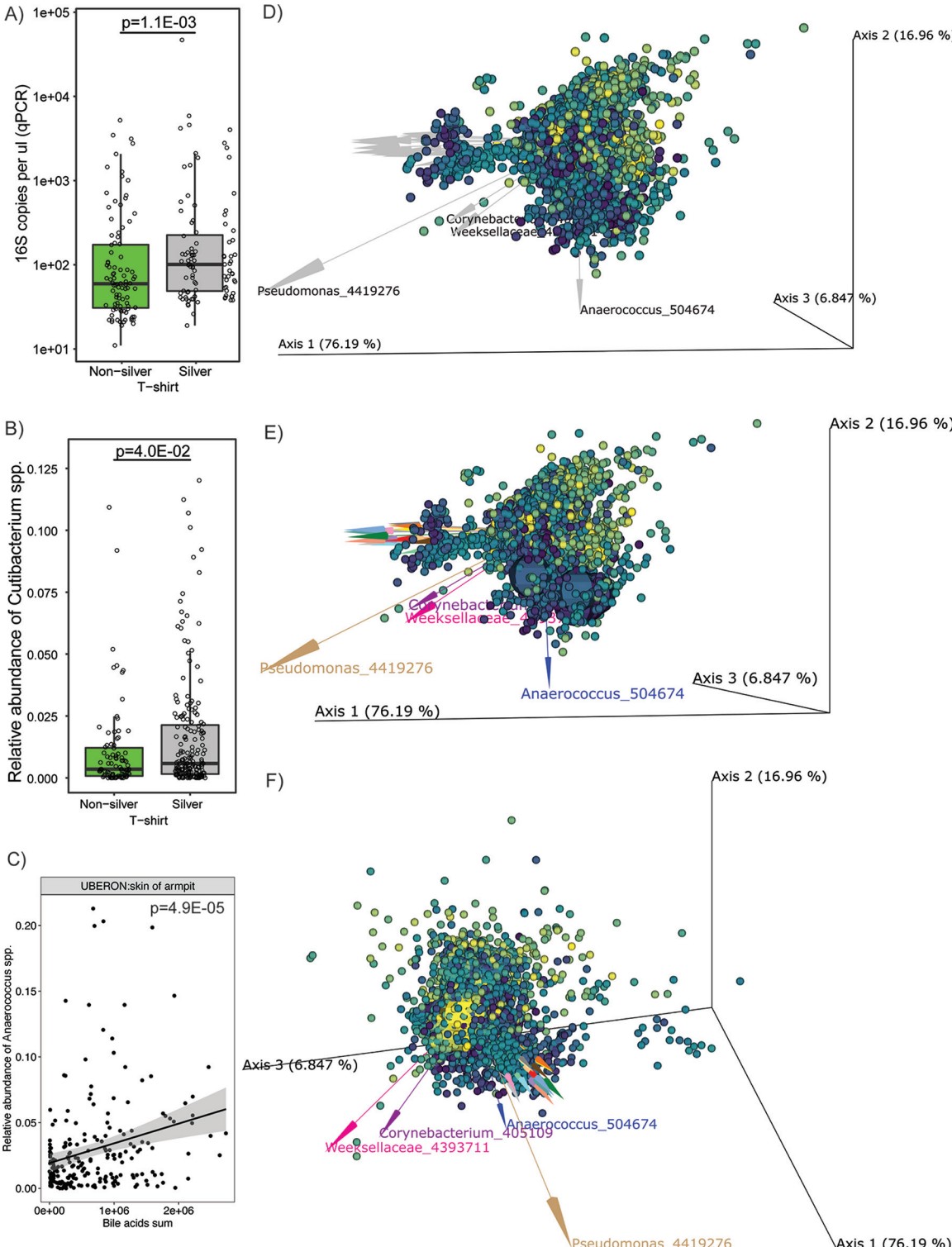

**FIG 3** Microbiome metabolome cooccurrence determined by mmvec. (A and B) qPCR (A) and bacterial Shannon diversity (B) results of all samples combined and stratified by the T-shirt phase, respectively. (C) *Cutibacterium* species distribution by the T-shirt phase. (D) mmvec biplot showing the metabolome, where each sphere is a metabolite feature that is color coded based on the enrichment in the silver phase, and the microbiome, where each arrow represents a microbe. A proximity of both metabolites and microbes is indicative of their cooccurrence. (E) Bile acids cooccur with *Anaerococcus* spp. Bile acids were generally decreased in the silver phase (dark large spheres). (F) MUFAs tend to cooccur with *Corynebacterium* spp. MUFAs were all enriched in the silver phase (light large spheres).

Spearman test) and the bacterial biomass on the skin of the back, suggesting that bacteria and their biomass are involved in the bioconversion of MUFAs.

The impact was the largest where the contact between silver clothes and skin was the highest: the upper back and chest skin. During the silver phase, the skin of the torso (chest and back) became enriched in *Lactobacillus*, *Anaerococcus*, *Gemellaceae*, *Cutibacterium*, and *Lactococcus* ($P < 0.05$, Mann-Whitney U test). The silver shirts themselves were enriched in *Cutibacterium* spp. (Fig. 3B), among some other taxa. Anecdotally, volunteer 11 mentioned incident upper back acne when the silver shirts were worn, which might suggest intermittent increases in the relative abundance of *Cutibacterium* spp. on the upper back.

When looking at the log abundance ratios, we noticed particular bacterial genera that were enriched on skin when wearing silver-threaded clothing, compared to nonsilver clothing. Bacterial genera that were consistently more abundant on the chest, upper back, and lower back when silver clothing was worn were *Bacteroides*, *Acinetobacter*, *Peptoniphilus*, *Akkermansia*, *Facklamia*, *Haemophilus*, *Helcococcus*, *Rothia*, *Lactococcus*, and *Erwinia* spp. (Songbird/Qurro results not shown). These small changes in microbial diversity have led to much more significant changes in the skin metabolome, discussed above.

Our results indicate that the skin microbiome is strongly determined by individuality, body site, and gender, as reported earlier (28, 29). *Staphylococcus* and *Corynebacterium* were the main bacterial groups in armpit, chest, and upper and lower back skin (Fig. S7). Females had a higher relative abundance of *Staphylococcus* while males had a higher relative abundance of *Corynebacterium*, as documented before (25). These bacterial groups were not significantly impacted by the silver-threaded textiles, and the proportions of the most abundant skin taxa did not change (Fig. S7a and b). The impact of silver-threaded shirts on the bacterial composition was subtle but noticeable. Silver increased the female skin microbial Shannon diversity significantly ($P = 0.036$, Mann-Whitney U test) (Fig. S1c), but the diversity was less so affected on the chest and upper back ($P = 0.05$, Mann-Whitney U test). This diversity difference was not seen on male skin ($P = 0.99$, Mann-Whitney U test) and is likely outweighed by the more abundant *Corynebacterium* (and *Staphylococcus*) spp. present on male skin (Fig. S7a). Notably, this increase in microbial diversity also corresponds to a decrease in the chemical diversity.

## DISCUSSION

Silver is the most commonly used antimicrobial agent for textile application, and silver nanoparticles (AgNP) are the most used nanoparticles in consumer products (30). However, there has been limited research on the physiological responses to such textiles by the skin microbiome and metabolome. In this study, we have found that the material used in clothing can cause changes in the chemistry and microbial community of the skin surface. The impact of clothing on skin chemistry and microbiome is highly individualized and differs for genders and body sites (28, 29). This also corresponds with findings from a study on the T-shirt microbiome which found the textile to be highly individual in microbial composition, diversity, and biomass (29). Although in this study, the abundant genera remained unaffected (Fig. S7a and b), low-abundant taxa were perturbed (Fig. S1d) and the use of antibacterial T-shirts also resulted in higher bacterial biomass on skin. Silver textiles can manipulate the microbiome and metabolome of the skin but do not reduce microbial biomass.

Silver-coated-yarn-threaded T-shirts had a small but significant impact on the skin microbiome (mainly for females), leading to a higher richness and diversity, including higher abundance of some odor-causing species, such as *Anaerococcus*. This difference between males and females could have been due to the slightly higher content of silver in the females' T-shirts. The amount of X-Static silver-threaded yarn was 5% for the female shirt and 4% for the male shirt. During the silver phase in male participants, a higher bacterial biomass was found, with *Staphylococcus* and *Corynebacterium* being the most dominant (Fig. S5d and Fig. S7). These results are counterintuitive and surprising, since the primary reason for incorporating silver ions into garments is to

manage or reduce the bacterial load and to control odor formation. This study shows the opposite outcome from the intended primary reason for including silver particles, as a higher bacterial load on skin was found, and particular malodor-associated taxa were increased on skin.

The altered skin microbiome corresponded with a large impact on the skin metabolome. The microbiome and metabolome data suggest that silver textiles alter the bacterial populations on the skin in a way that, most notably, results in the accumulation of monounsaturated fatty acids (MUFAs). These MUFAs were the key compounds on skin affected by the silver textiles (Fig. 2). All of these compounds changed in unison with the silver phase and were detected in similar abundances (Fig. 2D). The presence of odd-carbon short-chain fatty acids (OCS-FAs) among these discriminating compounds strongly suggests a microbial origin. Bacteria can produce odd-chain fatty acids related to the production of propionic acid (31–33). A multi-omics analysis suggested a link between the abundance of MUFAs, e.g., myristoleic acid, and certain microbes, such as *Corynebacterium* and *Cutibacterium* spp. (Fig. 3F and Fig. S6e). We hypothesize that the silver-threaded shirts led to a higher sebum production on the skin, which led to higher bacterial biomass and to certain species being enriched during the silver phase (Fig. S6b to d). Interestingly, this increase in microbial diversity also corresponds to a decrease in the chemical diversity (Fig. S7a). Yet, in other studies the impact of skin cosmetics and antimicrobial ingredients has led to both an increase in chemical diversity and an increase in microbial diversity (23).

Monounsaturated fatty acids are the major fatty acids that are present in human sebum (34, 35). Corynebacteria are lipophilic bacteria that rely primarily on fatty acids as a food source (27, 36, 37). A cooccurrence between MUFAs and corynebacteria could be due to the ability of the latter to produce FadD enzymes, which is the first step to break down fatty acids in the beta-oxidation pathway (38). In another example, *Cutibacterium acnes*, a Gram-positive skin commensal, has been identified to be a contributing factor to acne on a strain level (39). These species also have the enzymatic capacity to manipulate the MUFAs on the skin (40). *Cutibacterium* spp. were enriched on the skin of the torso, which corresponded with an increase in the production of medium-chain fatty acids, including odd-carbon ones (Fig. S6b). The silver textile shirts themselves were similarly enriched in *Cutibacterium* spp. (Fig. S6f) and contained fewer staphylococci. Further exploration of microbial involvement in the bioconversion of sebum into MUFAs is warranted. The results further showed a separation of textiles and skin metabolomes and showed that clothes differ in this respect from the skin sites (Fig. S3). Other studies have shown composition differences in axillary and textile microbiomes, and similarly, we saw that the skin and textile metabolomes are behaving independently of each other (2).

This study also identified a range of different primary bile acids, which were solely found in the armpits of participants. The mmvec analysis established a clustering of the bile acids in both the silver and nonsilver phases. Bile acids may help in solubilizing underarm lipids, which can help reduce friction in the underarm. In our study, these bile acids were associated and correlated with *Anaerococcus* species (Fig. 3e). *Anaerococcus* is a low-abundant species in the underarm and associated with higher malodor scores (21). Bile acids are host produced by the apocrine sweat glands (41) and were detected only in the underarms (23). Bile acids have an important role in lipid metabolism in the gut to manage the microbial community; therefore, they have a therapeutic role against pathogens (42–44). However, the function of bile acids on the skin remains largely unknown (42). The results of a study found certain probiotic bacteria, including *Lactobacillus*, *Bifidobacterium*, and *Bacillus*, to be resistant to bile acids, thus reducing bile acids' relative antimicrobial activity (45, 46). This suggests the selective capabilities of bile acids on certain strains of bacteria. In this paper, the silver textile has caused slight distortion in the microbial community, which has led to these larger changes in metabolism on the skin. This highlights the importance of this kind of microbiome/metabolome study when developing a functional textile.

**Conclusion.** In this study, we investigated the impact of clothes with antimicrobial properties on the skin by studying induced changes of the microbiome and metabolome. The main changes to the skin microbiome and metabolome include an increase of bacterial biomass and an increase of monounsaturated fatty acids in the silver phase. The observation of bile acids on the skin was found in the silver and nonsilver phases. Bile acids have an important role in lipid metabolism in the gut, but their function on the skin remains unknown. This study indicates that the microbiome and metabolome are interlinked, and the textiles did indeed cause changes on the skin microbiome which drove significant chemical changes on the skin due to silver antimicrobials. Textile and its active ingredients do have an impact on human skin biology and chemistry and may provide a direct way to manipulate skin chemistry for health and wellbeing. However, more extensive research is needed on more volunteers into the strain significance of the microbiome changes and the exact origin of the chemical changes on the skin.

## MATERIALS AND METHODS

**Study design.** Healthy volunteers were enrolled in this study with one visit every week for 11 weeks. Personal care products and laundry detergent were provided to all volunteers used in the study. Showering during the 11-week study was allowed only with the shampoo, soap, and deodorant that were provided. During the washout period, the first 2 weeks, participants asked to wear the non-X-Static shirt. During the first 3 weeks, post-washout period, participants were asked to wear the X-Static shirt, followed by wearing the non-X-Static shirt from week 4 to week 6, and finishing with the X-Static yarn-threaded T-shirt again. X-Static yarn is treated with silver to inhibit the growth of bacteria on fabrics, eliminating human-based odor (https://noblebiomaterials.com/x-static/). The amount of X-Static yarn was 5% for the female shirt and 4% for the male shirt. Non-X-Static shirts were not treated with silver. Participants were asked to take no shower 24 h before sample collection. Participants all lived in the temperate climate and region of San Diego, CA, during the time of study (April to June 2017). The study was reviewed and approved by the University of California San Diego (UCSD) Institutional Review Board under the identifier (ID) 161694 on 3 November 2016. Sample collection, preparation, and data acquisition were performed in the Skaggs School of Pharmacy of UCSD for metabolomics samples and in the Department of Pediatrics and Computer Science & Engineering of UCSD for microbiome samples.

**Sample collection.** A sterile swab, either with alcohol solution or with saline solution, was used to collect samples from small skin areas (2 by 2 in.) by swabbing the skin surface for approximately 10 s. Two samples from 5 body sites were collected for a total of 10 samples per volunteer per week. The samples were collected from the right armpit, left armpit, upper back, lower back, and chest. After collection, the swabs were put in a 96-well plate (one for metabolite and one for sequencing), containing the appropriate extraction buffer.

**DNA extraction and sequencing.** 16S rRNA gene amplicon sequencing was performed following the Earth Microbiome Project protocols (47), as described before (42). Briefly, DNA was extracted using the MoBio PowerMag Soil DNA isolation kit, and the V4 region of the 16S rRNA gene was amplified using barcoded primers (48). PCR was performed in triplicate for each sample, and V4 paired-end sequencing (48) was performed using Illumina HiSeq (La Jolla, CA, USA). Raw sequence reads were demultiplexed and quality controlled using the defaults, as provided by QIIME 1.9.1 (49). The primary operational taxonomic unit (OTU) table was generated using Qiita (https://qiita.ucsd.edu/), using the UCLUST (50) closed-reference OTU picking method against the Greengenes 13.5 database (51). Sequences can be found in EBI under accession number EBI ERP138010 or in Qiita (https://qiita.ucsd.edu) under study ID 11272.

**qPCR.** We analyzed the absolute quantity of bacteria on a subset of the samples, using quantitative PCR (qPCR). A total of 192 samples were processed for qPCR 16S rRNA gene quantitation which included 168 primary samples and 24 DNA extraction blanks as reference. Specifically, 1 $\mu$L of neat genomic DNA (gDNA) from each of the 12 volunteers at week 8 (nonsilver phase) and week 11 (silver T-shirt phase) along with each sample site including right armpit, left armpit, chest, lower back, upper back, left armpit clothing, and right armpit clothing was amplified in a 10-$\mu$L PCR mixture with DyNAmo HS SYBR green master mix (ThermoFisher catalog no. F410L) in triplicate for qPCR analysis on the Roche LightCycler 480 using the same amplification conditions and primers (16S primers 515f and 806rb) as described for the microbiome analysis. An isolate of *Vibrio fischeri* ES114 with known genome size and gDNA concentration, determined with a Qubit fluorometer (ThermoFisher), was 10-fold serially diluted six times (135,000 to 1.35 genome copies) and used as a positive control. The qPCR amplification efficiency was 95.68% and 93.34% for the two qPCR runs. The conservative level of detection was 135 copies for each run with the $R^2$ being 0.99749 and 0.99724 for the two runs. Reported values are in 16S rRNA gene copies per microliter. Since each gDNA extraction was 100 $\mu$L, one could multiply by 100 to indicate total 16S RNA gene copies per DNA extraction, but we have chosen to leave this in the original form of copies per microliter.

**Metabolite extraction and UPLC-quadrupole time of flight MS/MS analysis.** Skin swabs were extracted and analyzed using a previously validated workflow described in reference 52. All samples were extracted in 200 $\mu$L of 50:50 ethanol-water solution for 2 h on ice and then overnight at −20°C. Swab sample extractions were dried down in a centrifugal evaporator and then resuspended by vortexing and sonication in a 100-$\mu$L 50:50 ethanol-water solution containing two internal standards (ISTDs). The ethanol-water extracts were then analyzed using an ultraperformance liquid chromatography (UPLC)-MS/MS method described in reference 42. A ThermoScientific Dionex 3000 UPLC for liquid

chromatography and a Maxis Impact II mass spectrometer (Bruker Daltonics), controlled by the software packages (Bruker Daltonics) and equipped with an electrospray ionization (ESI) source, were used. UPLC conditions of analysis were an 1.7-$\mu$m C$_{18}$ (50- by 2.1-mm) ultrahigh-performance liquid chromatography (UHPLC) column (Phenomenex), column temperature of 40°C, flow rate of 0.5 mL/min, mobile phase A of 99.9 water-0.1 formic acid (vol/vol), and mobile phase B of 99.9 acetonitrile-0.1 formic acid (vol/vol). A linear gradient was used for the chromatographic separation: 0 to 2 min, 0 to 20% B; 2 to 8 min, 20 to 99% B; 8 to 9 min, 99 to 99% B; 9 to 10 min, 0% B. Full-scan MS spectra ($m/z$ 80 to 2,000) were acquired in a data-dependent positive ion mode. Instrument parameters were set as follows: nebulizer gas (nitrogen) pressure, $2 \times 10^5$ Pa; capillary voltage, 4,500 V; ion source temperature, 180°C; dry gas flow, 9 L/min; and spectrum rate acquisition, 10 spectra/s. MS/MS fragmentation of the 10 most intense selected ions per spectrum was performed using ramped collision-induced dissociation energy, ranging from 10 to 50 eV to get diverse fragmentation patterns. MS/MS active exclusion was set after 4 spectra and released after 30 s. Mass spectrometry data for this study can be found at MSV000081379.

**LC-MS data processing and analysis.** LC-MS raw data files were converted to mzXML format using msConvert (ProteoWizard). MS1 features were selected for all LC-MS data sets collected using the open-source software MZmine 2 (53) with the following parameters: mass detection noise level was 1,000 counts, chromatograms were built over a 0.01-min minimum time span, with 3,000-count minimum peak height and 20-ppm mass tolerance, features were deisotoped and aligned with 20-ppm tolerance and 0.1-min retention time tolerance, and aligned features were filtered based on a minimum 3-peak presence in samples and based on containing at least 2 isotopes. Subsequent blank filtering, total ion current, and internal standard normalization were performed for representation of relative abundance of molecular features and for principal-coordinate analysis (PCoA). A Kruskal-Wallis test was used to find differentially abundant metabolites across left and right armpits, and $P$ values were adjusted for multiple hypothesis testing using the false-discovery rate (FDR) method. To visualize different distributions of metabolites across left and right armpits, a heat map of the differentially abundant metabolites (FDR-adjusted $P$ value of <0.05) was created using the Complex Heatmap package version 2.8.0 (54) in R (55). Rows and columns were clustered using the Euclidean distance and complete clustering method. Only differentially abundant metabolites with a putative class annotation were displayed in the heat map.

**Metabolite annotation.** MZmine-preprocessed MS/MS fragmentation spectra were submitted to feature-based mass spectral molecular networking through the Global Natural Products Social Molecular Networking Platform (GNPS) (22) and searched against all GNPS spectral libraries. The exact mass and MS/MS spectral matching are equivalent to the level 2 identification according to paragraph 2.9 of "Proposed minimum reporting standards for chemical analysis Chemical Analysis Working Group (CAWG) Metabolomics Standards Initiative (MSI)" (56). To further enhance chemical structural information, we performed *in silico* structure annotation using Network Annotation Propagation and created consensus chemical classes per molecular family using the GNPS MolNet Enhancer workflow (https://ccms-ucsd.github.io/GNPSDocumentation/molnetenhancer/) (57) with chemical class annotations retrieved from the ClassyFire chemical ontology (57, 58). The GNPS molecular networking job can be accessed at https://gnps.ucsd.edu/ProteoSAFe/status.jsp?task=dd4ed24be55b4e0c96d542a12b7b464e; *in silico* Network Annotation Propagation results can be accessed at https://proteomics2.ucsd.edu/ProteoSAFe/status.jsp?task=4cd9cd5b7f94463abc8edd4aca94c415, (59), and MolNetEnhancer results can be accessed at https://gnps.ucsd.edu/ProteoSAFe/status.jsp?task=d341e23d0aa8489d9776118c48db5a67.

**Microbiome data analysis.** All sequence data were quality filtered to discard sequences with a quality score of <20. OTUs were assigned taxonomy using the Greengenes (v13_8) reference database. Samples were rarefied to 2,580 sequences per sample, and alpha diversity for each sample and distances between samples were calculated using QIIME v1.9.1 (49). Pairwise differences in alpha diversity were tested using the nonparametric Wilcoxon tests. Taxonomy abundance log-fold differentials were calculated through QIIME2 using the Songbird plug-in (60) with visualization using Qurro (60, 61).

**Random forest analysis.** A random forest classification model (62) was used to identify microbes and metabolites that separated the textile phases. This model was run using the randomForest package in R with 5,000 trees and 59 variables tried at each split and with stratification due to differential sample numbers in each disease class.

**LEfSe.** We used the linear discriminant analysis (LDA) effect size (LEfSe) method (http://huttenhower.sph.harvard.edu/lefse/) (63) for microbe and metabolite biomarker discovery, which performs a combined assessment of statistical significance and biological relevance. The tool utilizes a nonparametric Kruskal-Wallis test to investigate group differences, using a sample-wise normalized matrix of relative abundances, and determines the effect size of a given taxon using LDA. We performed this analysis using the default settings (alpha = 0.05, effect-size threshold of 2).

**mmvec.** Microbe-metabolite cooccurrence probabilities were calculated using mmvec, a neural network approach trained to predict metabolite abundances given the presence of a single microbe (27). This model was trained using three principal axes with a batch size of 10,000 and 10,000 epochs. mmvec performs cross-validation by evaluating how well the metabolites can be predicted solely from the microbe abundances in the samples.

**qPCR statistics.** Sample sites were compared using a nonparametric Kruskal-Wallis test with multiple comparisons applying the Benjamini-Hochberg FDR. To compare the specific body sites from the effects of silver T-shirt wear, a one-tailed Wilcoxon matched-pair signed-rank test was used to compare statistical differences between uses of a silver T-shirt across body sites (paired data per individual).

## SUPPLEMENTAL MATERIAL

Supplemental material is available online only.

**FIG S1**, TIF file, 1.5 MB.
**FIG S2**, TIF file, 1.6 MB.
**FIG S3**, TIF file, 2.2 MB.
**FIG S4**, TIF file, 1.5 MB.
**FIG S5**, TIF file, 1.4 MB.
**FIG S6**, TIF file, 2 MB.
**FIG S7**, TIF file, 1.8 MB.
**FIG S8**, TIF file, 1.3 MB.
**FIG S9**, TIF file, 1.9 MB.
**FIG S10**, TIF file, 1.2 MB.

## ACKNOWLEDGMENTS

K.D. and P.C.D., study and experimental design; K.D., experimental organization and setup; K.D. and C.C., metabolite and microbial sample collection; A.V.M. and A.A.A., mass spectrometry data collection; A.V.M. and A.A.A., mass spectrometry data analysis; G.H., 16S rRNA sequencing; C.C. and G.A., metadata organization; C.C. and J.J.M., microbial data analysis; M.E., heat map of chemical classes. A.V.M., A.A.A., C.C., K.D., P.C.D., R.K., and R.B. wrote the manuscript. All authors critically revised the manuscript for important intellectual content.

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
