## [Reviewer comments · mSystems]

The molecular effect of wearing silver-threaded clothing on the human skin

Alexey Melnik, Chris Callewaert, Kathleen Dorrestein, Rosie Broadhead, Jeremiah Minich, Madeleine Ernst, Greg Humphrey, Gail Ackermann, Rob Gathercole, Alexander Aksenov, Rob Knight, and Pieter Dorrestein

Corresponding Author(s): Pieter Dorrestein, University of California, San Diego

Review Timeline:

Submission Date:	September 21, 2022
Editorial Decision:	October 24, 2022
Revision Received:	December 8, 2022
Accepted:	December 16, 2022

Editor: Carla Porto

Reviewer(s): Disclosure of reviewer identity is with reference to reviewer comments included in decision letter(s). The following individuals involved in review of your submission have agreed to reveal their identity: Rita de Cassia Pessotti (Reviewer #1); Celio Fernando Figueiredo Angolini (Reviewer #2)

Transaction Report:

DOI: <https://doi.org/10.1128/msystems.00922-22>

October 24, 2022

Dr. Pieter C. Dorrestein
University of California, San Diego
Department of Pharmacology, Chemistry and Biochemistry
Skaggs School of Pharmacy and Pharmaceutical Sciences
La Jolla, CA 92093

Re: mSystems00922-22 (The molecular effect of wearing silver-threaded clothing on the human skin)

Dear Dr. Pieter C. Dorrestein:

Thank you for submitting your manuscript to mSystems. We have completed our review and I am pleased to inform you that, in principle, we expect to accept it for publication in mSystems. However, acceptance will not be final until you have adequately addressed the reviewer comments.

Please include in the text the consent form obtained for studies in humans.

Thank you for the privilege of reviewing your work. Congratulations to Alexey and all team for the great work. Below you will find instructions from the mSystems editorial office and comments generated during the review.

Preparing Revision Guidelines

Congratulations for the All the best

Carla Porto

Editor, mSystems

Journals Department
Reviewer comments:

Reviewer #1 (Comments for the Author):

I would like to suggest that next time authors add line numbers to the text to facilitate the review process.

Reviewer #2 (Comments for the Author):

This is a very interesting work that brings some information about the composition of the skin microbiota under the effect of fabrics containing silver with antimicrobial claims. The experimental design was well executed and the results were quite promising, but I had some questions about the discussions.

Regarding chemical diversity, it was not clear to me how this result was obtained, I understood that the Shannon diversity index was used, but exactly how was the index calculated? For example, looking at figure 1 a) I would expect a greater number of only green nodes, showing that there is a greater number of molecules without the presence of silver. But what I see seems to be the opposite, the difference is small, but a greater amount of only grey nodes are observed, which would indicate a greater production of different molecules in the presence of silver, correct?

The other observations I found all very interesting and consistent with the data. Just out of curiosity, did you investigate other influences such as climate, which could be equally affecting all the volunteers in the different periods (with silver and without silver clothes). Because as the two periods (groups) were run at different times, maybe the results could be masked by a variable that affects everyone equally.

Did you collect sensory information from the participants? If they noticed a difference in odours, skin appearance, etc.

Did you do any evaluation of volatile compounds?

Summary

This manuscript, entitled “The molecular effect of wearing silver-threaded clothing on the human skin”, presents interesting and novel information that is of interest of mSystems and its audience. They investigated the impact of clothes with silver threading on the chemistry and microbiome of the human skin. Silver-threaded clothing are often wore by athletes to improve hygiene. Counterintuitively, the authors observed an increase on microbial biomass on skin when wearing this kind of clothing. Even though the most abundant taxa were not affected, an increase in diversity and richness of low-abundant bacteria was observed. Chemical diversity was decreased, and an increase in the abundance of various monounsaturated fatty acids (MUFAs) was observed. Notably, results varied among volunteers and body sites, which major changes being mainly observed for the female group and upper back. This study contributes to the understanding of how textile influence our skin metabolome and microbiome, raising the awareness on the impact of our daily choices of clothing on skin health.

Overall, the manuscript is well-written. The authors planned well their experiments, used appropriated controls, and did a good and deep statistical analysis of their data. A limitation of this study is that a high variability among volunteers was observed, which is expected, and they acknowledged that further studies with a higher number of volunteers is needed to better understand this topic.

I have some minor concerns, shown below. I would like to suggest that next time authors add line numbers to the text to facilitate the review process.

Minor concerns

- As stated in the manuscript, there was variability among volunteers in the biological and chemical response to wearing clothes with silver threading. Nobody has identical microbiomes, and this natural variability from person to person might be in part responsible for the variability observed in the outcomes of this study. With this in mind, it would be interesting to add to the supplemental information a comparison of the microbiome at time zero among volunteers. How different are they? What are the major differences? Maybe this information could be useful to understanding the variability in the outcomes after wearing silver threading.

- In the methods section it is disclosed that females wore clothes with a higher silver content (5% versus 4% in male clothes), and this information is not presented in any other section. The study showed that changes were more pronounced in the females group. Maybe a higher silver content could be one of the factors driving the observed difference. Authors should acknowledge this in the text.

- It is mentioned in the manuscript that silver threading is often wore by athletes to control odour and improve hygiene due to its antimicrobial property. However, the results show the opposite – a higher bacterial biomass was observed, as well as a higher abundance of some odour-causing species, such as *Anaerococcus*. It would be interesting to add this perspective to the discussion, that the silver threading is not behaving as it is expected – it is actually doing the opposite. This highlights the importance of this kind of microbiome/metabolome study when developing a functional textile.

- The defined noise level (during mass spec data processing) seems very low (1000). Were the signals in general at low abundance? Why such a low noise level was chosen?

- What is the confidence level of the annotated compounds? How were they annotated? Please add this information to the text.

- **Page 6, last sentence of the second paragraph:** where it says “Figure 1a”, it is actually 1b, and vice-versa.

- **Page 7, Figure 1:** please add to the figure legend the meaning of: size of the nodes, edge thickness, and numbers inside the nodes.

- **Page 9, figure 2a and 2b:** I’m not convinced by these plots that silver and non-silver groups are different in the upper and lower back. I don’t see any separation of these groups in these 2 PCoA plots, which is confusing given that the text says “When considering body parts separately, the effects of silver could be more clearly observed”. I would like to ask the authors to please elaborate more the discussion of these plots.

- **Page 10, second last line:** “Corynebacterium spp. Were found to be one of the species that tend to co-occur with”. Typo: “Were” should be written in lowercase.

- **Page 15:** where it says “Supplementary Figure 8f”, authors probably meant “Supplementary Figure 6f” since there is no supp. Fig. 8f.

- **Page 16, “Study Design” topic, line 12:** there is a typo: “Srudy”. In the same paragraph, the period is missing at the end of the paragraph.

- **Figures that are not cited in the text:** Supplementary Figure 5a and 5c, Supplementary Figure 8a, 8b, 8d. Please cite them.

- **Supplementary Figure 2:** Please add to the figure legend what the numbers on the top of the box plots mean, and standardize their display (they all should have the same number of decimal places).

- **Supplementary Figure 2B:** I suggest that authors put right and left armpit box plots side by side, lower and upper back as well.

- **Supplementary Figure 3 and 4:** which time point(s) these plots and heatmap represent? Add this information to the figure legend.

- **Supplementary Figure 3C:** It would be clearer if authors write the full information in the legend, e.g., 3C: PCA highlighting only armpit samples and colored based on volunteer (like A right).

- **Supplementary Figure 4:** There are some colors in the color-code chart that I didn’t find in the heatmap, like lower back and blank. If they are indeed not present, please update the color-code accordingly. However, I suggest that it is only kept in the figure samples that are from armpits, since the point of this figure is to showcase the chemical differences between right and left armpit: “*A number of compounds that may be linked to bacterial origin - acylcarnitines, phospholipids and bile acids have been found to have different abundances in two armpits*”. The way this figure is presented makes it hard to see this point: there’s no explanation about tree diagrams on top and right side of the figure, and it is not described how

they were constructed. Please add this information to the figure legend and/or methods section. I can't understand this "Color key and Histogram". What is "value" and "count"? The "count" axis goes from 0-80000, but I don't see anything in the plot, I guess the histogram is missing in this plot?

- **Supplementary Figure 5A:** add "(qPCR)" to the axis title: "16S copies per uL (qPCR)".
- **Supplementary Figure 5B:** write in the caption what C, UB, LB, CLA, CRA stands for.
- **Supplementary Figure 6:** in the legend for letter D *Cutibacterium* is not in italic. There is a typo in letter F: "relativa".
- **Supplementary Figures 6D, 8B and 8D:** I strongly suggest that authors change the red/green combination to a color combination that is colorblind-friendly.
- **Supplementary Figure 8B:** what does "personal" mean?
- **Supplementary Figure 9:** standardize the display of p-values to either scientific format or decimal form.
- **Supplementary Figure 10D:** what is the timepoint of this comparison between armpit side?

Reviewer #1 (Comments for the Author):

I would like to suggest that next time authors add line numbers to the text to facilitate the review process.

A: We appologize for the inconvenience regarding this issue. We would definitely consider this for the next submission

Summary

This manuscript, entitled “The molecular effect of wearing silver-threaded clothing on the human skin”, presents interesting and novel information that is of interest of mSystems and its audience. They investigated the impact of clothes with silver threading on the chemistry and microbiome of the human skin. Silver-threaded clothing are often wore by athletes to improve hygiene. Counterintuitively, the authors observed an increase in microbial biomass on skin when wearing this kind of clothing. Even though the most abundant taxa were not affected, an increase in diversity and richness of low-abundant bacteria was observed. Chemical diversity was decreased, and an increase in the abundance of various monounsaturated fatty acids (MUFAs) was observed. Notably, results varied among volunteers and body sites, which major changes being mainly observed for the female group and upper back. This study contributes to the understanding of how textile influence our skin metabolome and microbiome, raising the awareness on the impact of our daily choices of clothing on skin health. Overall, the manuscript is well-written. The authors planned well their experiments, used appropriated controls, and did a good and deep statistical analysis of their data. A limitation of this study is that a high variability among volunteers was observed, which is expected, and they acknowledged that further studies with a higher number of volunteers is needed to better understand this topic. I have some minor concerns, shown below. I would like to suggest that next time authors add line numbers to the text to facilitate the review process.

Minor concerns

- As stated in the manuscript, there was variability among volunteers in the biological and chemical response to wearing clothes with silver threading. Nobody has identical microbiomes, and this natural variability from person to person might be in part responsible for the variability observed in the outcomes of this study. With this in mind, it would be interesting to add to the supplemental information a comparison of the microbiome at time zero among volunteers. How different are they? What are the major differences? Maybe this information could be useful to understanding the variability in the outcomes after wearing silver threading.

A: Thank you. There is indeed a certain interindividual variability we need to take into account. We agree with the referee that it would be useful to add a graph with microbiome compositions on timepoint 0 in supplemental information. We have added this information as Supplemental Figure 10. It is also referred to in the main manuscript in the microbiome part of the Results section. It reads: “The high interindividual response in skin microbiome was further backed by the initial individual skin microbiome differences (**Supplementary Figure 10**).”

- In the methods section it is disclosed that females wore clothes with a higher silver content (5% versus 4% in male clothes), and this information is not presented in any other section. The study showed that changes were more pronounced in the females group. Maybe a higher silver content could be one of the factors driving the observed difference. Authors should acknowledge this in the text.

R: Thank you, I have added this point on the differences in silver content between males and female t-shirts in the results and discussion to read as follows:

“This difference between males and females could have been due to the slightly higher content of silver in the females t-shirts. The amount of x-static silver threaded yarn was 5% for the female shirt and 4% for the male shirt.”

- It is mentioned in the manuscript that silver threading is often worn by athletes to control odour and improve hygiene due to its antimicrobial property. However, the results show the opposite – a higher bacterial biomass was observed, as well as a higher abundance of some odour-causing species, such as *Anaerococcus*. It would be interesting to add this perspective to the discussion, that the silver threading is not behaving as it is expected – it is actually doing the opposite. This highlights the importance of this kind of microbiome/metabolome study when developing a functional textile.

A: Very good point. We have clarified these surprising counter-intuitive findings in the discussion section.

The second paragraph of the discussion section now reads as follows:

“Silver-coated yarn threaded T-shirts had a small but significant impact on the skin microbiome (mainly for females), leading to a higher richness and diversity including higher abundance of some odor-causing species, such as *Anaerococcus*. During the silver phase in male participants, a higher bacterial biomass was found with *Staphylococcus* and *Corynebacterium* being the most dominant (Supplementary Figure 5d and 7). These results are counter-intuitive and surprising, since the primary reason for including silver-ions into garments is to manage or reduce the bacterial load and to control odor formation. This study shows the opposite outcome for the intended primary reason for including silver particles, as a higher bacterial load on skin was found, and particular malodor-associated taxa were increased on skin.”

The last paragraph of the discussion now reads:

“In this paper the silver textile has caused slight distortion in the microbial community, which has led to these larger changes in metabolism on the skin. This highlights the importance of this kind of microbiome/metabolome study when developing a functional textile.”

(We couldn't have said it better)

- The defined noise level (during mass spec data processing) seems very low (1000). Were the signals in general at low abundance? Why such a low noise level was chosen?

A: During data acquisition we didn't want to omit any signal to construct reliable chromatograms. We did increase the threshold in feature finding to 3000 to filter out any noise.

- What is the confidence level of the annotated compounds? How were they annotated? Please add this information to the text.

A: We have added a new paragraph to method section called "Metabolite annotation":
MZmine preprocessed MS/MS fragmentation spectra were submitted to feature-based mass spectral molecular networking through the Global Natural Products Social Molecular Networking Platform (GNPS) [refs] and searched against all GNPS spectral libraries. The exact mass and MS/MS spectral matching is equivalent to the level 2 identification according to the paragraph 2.9 of "Proposed minimum reporting standards for chemical analysis Chemical Analysis Working Group (CAWG) Metabolomics Standards Initiative (MSI)"(<https://www.ncbi.nlm.nih.gov/pmc/articles/PMC3772505/>) To further enhance chemical structural information we performed in silico structure annotation using Network Annotation Propagation [ref] and created consensus chemical classes per molecular family using the GNPS MolNetEnhancer workflow (<https://ccms-ucsd.github.io/GNPSDocumentation/molnetenhancer/>) [ref] with chemical class annotations retrieved from the ClassyFire chemical ontology [ref]. The GNPS molecular networking job can be accessed at [https://gnps.ucsd.edu/ProteoSAFe/status.jsp?task=dd4ed24be55b4e0c96d542a12b7b464e;in-silico Network Annotation Propagation results](https://gnps.ucsd.edu/ProteoSAFe/status.jsp?task=dd4ed24be55b4e0c96d542a12b7b464e;in-silico%20Network%20Annotation%20Propagation%20results) at <https://proteomics2.ucsd.edu/ProteoSAFe/status.jsp?task=4cd9cd5b7f94463abc8edd4aca94c415> and MolNetEnhancer results at <https://gnps.ucsd.edu/ProteoSAFe/status.jsp?task=d341e23d0aa8489d9776118c48db5a67>.

- Page 6, last sentence of the second paragraph: where it says "Figure 1a", it is actually 1b, and vice versa.

A: Thank you for catching this, the last sentence of the second paragraph now reads:
"T-shirt phase (**Figure 1a**) or different body parts (**Figure 1b**) gives visualizations of associated molecular distributions."

- Page 7, Figure 1: please add to the figure legend the meaning of: size of the nodes, edge thickness, and numbers inside the nodes.

A: We have added the discretion to the legend which now reads:
Figure 1. Molecular networks for exploration of metabolome changes induced by silver fabric. Global molecular networks of metabolomics data color-coded by a) T-shirt phase and b) body part. c) The plot showing the ratio of abundances of metabolites during the silver versus non-silver phases on upper and lower back. Ratios higher and lower than one (i.e. molecules that increased and decreased in abundance) are highlighted in red and blue, respectively. Examples of some of the annotated molecules with the corresponding clusters from the molecular network that contain them are shown: monounsaturated fatty acids; bile acids and a surfactant. Clusters are colored the same way as in panel a). Numbers inside cluster nodes denote *m/z* as measured by a mass spectrometer, while size of the nodes is determined by the peak area under the curve. Thickness of the edges connecting nodes increases continuously with cosine similarity score increase. Consistent ratios of MUFAs' abundances across the cluster are suggestive of the same chemical forces responsible for the changes in their differences in silver vs. non-silver samples. The depicted compounds are highlighted by square nodes.

- Page 9, figure 2a and 2b: I'm not convinced by these plots that silver and non-silver groups are different in the upper and lower back. I don't see any separation of these groups in these 2 PCoA plots, which is confusing given that the text says "When considering body parts

separately, the effects of silver could be more clearly observed". I would like to ask the authors to please elaborate more the discussion of these plots.

A: We have clarified the statement to read as follows:

When considering body parts separately in unsupervised analysis, the effects of silver cannot be clearly observed: **Figure 2a** and **2b** shows PCoA plots for the upper and lower back samples with some separation of samples based on silver and non-silver groups. **Figure 2c** shows the volcano plot for these samples. Several features appeared to be significant in discriminating silver and non-silver sample groups. The features were predominantly attributed to a single network cluster shown on **Figure 1c**, indicating their structural similarity

- Page 10, second last line: "Corynebacterium spp. Were found to be one of the species that tend to co-occur with". Typo: "Were" should be written in lowercase.

A: Thank you for catching this typo. Adjusted.

- Page 15: where it says "Supplementary Figure 8f", authors probably meant "Supplementary Figure 6f" since there is no supp. Fig. 8f.

A: Thank you - adjusted.

- Page 16, "Study Design" topic, line 12: there is a typo: "Study". In the same paragraph, the period is missing at the end of the paragraph.

A: Thank you for catching this typo. Adjusted.

- Figures that are not cited in the text: Supplementary Figure 5a and 5c, Supplementary Figure 8a, 8b, 8d. Please cite them.

A: Thank you - adjusted in the main manuscript. R: Supplementary Figure 8 a,b,d are cited in microbiome results and discussion. Supplementary Figure 5a and c are cited in microbiome results.

- Supplementary Figure 2: Please add to the figure legend what the numbers on the top of the box plots mean, and standardize their display (they all should have the same number of decimal places).

A: Supplementary Figure 2 is now adjusted to show p-values in the same scientific notation with the same number of significant figures.

The legend is also adjusted and now reads:

“Mass spectrometry chemical diversity separated by silver vs non-silver t-shirt phase. Shannon diversity calculated for and combined by: a) gender b) body part sampled c) volunteer assigned number. P-value of Shannon diversity measure is placed above each box plot”

- Supplementary Figure 2B: I suggest that authors put right and left armpit box plots side by side, lower and upper back as well.

A: This has been adjusted, thank you

- Supplementary Figure 3 and 4: which time point(s) these plots and heatmap represent? Add this information to the figure legend.

A: Legends are now read:

Supplementary Figure 3

General characterization of metabolomics data. a) Principal coordinate analysis colored by the two most prevalent separation based on body site (left) and volunteer sampled (right) for all samples at all time points. b) Principal coordinate analysis highlighting only armpit samples at all time points and colored based on which armpit side of the body was sampled like a) on the left. Enlarged spheres show samples from t-shirt material adjacent to armpits. c) Principal coordinate analysis plot colored by the volunteer samples the same as in a) on the right. Several volunteers' armpit samples express distinct clustering between actual armpits and clothing adjacent to that armpit.

Supplementary Figure 4

Heatmap of metabolites from armpits that have different distributions within left and right armpit samples of all volunteers combined at all timepoints. Chemical classes are putative estimates retrieved through the MolNetEnhancer⁵⁸ workflow and based on similarities in MS2 fragmentation patterns.

- Supplementary Figure 3C: It would be clearer if authors write the full information in the legend, e.g., 3C: PCA highlighting only armpit samples and colored based on volunteer (like A right).

A: Legend are now reads:

Supplementary Figure 3

General characterization of metabolomics data. a) Principal coordinate analysis colored by the two most prevalent separation based on body site (left) and volunteer sampled (right) for all samples at all time points. b) Principal coordinate analysis highlighting only armpit samples at all time points and colored based on which armpit side of the body was sampled like a) on the left. Enlarged spheres show samples from t-shirt material adjacent to armpits. c) Principal coordinate analysis plot colored by the volunteer samples the same as in a) on the right. Several volunteers' armpit samples express distinct clustering between actual armpits and clothing adjacent to that armpit.

- Supplementary Figure 4: There are some colors in the color-code chart that I didn't find in the heatmap, like lower back and blank. If they are indeed not present, please update the color-code accordingly. However, I suggest that it is only kept in the figure samples that are from armpits, since the point of this figure is to showcase the chemical differences between right and left armpit: "A number of compounds that may be linked to bacterial origin - acylcarnitines, phospholipids and bile acids have been found to have different abundances in two armpits". The way this figure is presented makes it hard to see this point: there's no explanation about tree diagrams on top and right side of the figure, and it is not described how they were constructed. Please add this information to the figure legend and/or methods section. I can't understand this "Color key and Histogram". What is "value" and "count"? The "count" axis goes from 0-80000, but I don't see anything in the plot, I guess the histogram is missing in this plot?

A: We thank the reviewer for this constructive feedback and agree that the main findings were not well presented in the heatmap. We have therefore revised the heatmap, so that patterns of differentially abundant features across left and right armpits and different putative chemical classes become more apparent. We have reduced the heatmap so that it only contains data from left and right armpits as suggested by the reviewer. Furthermore, we limited the amount of metabolite features displayed to only features, which showed significantly differentially abundant (Kruskal-Wallis, FDR-adjusted $P < 0.05$) across left and right armpits:

Supplementary Figure 4

Heatmap of metabolites from armpits that have different distributions within left and right armpit samples of all volunteers combined at all time points. All features displayed are significantly differentially abundant (Kruskal-Wallis, FDR-adjusted $P < 0.05$) across left and right armpits. Chemical classes are putative estimates retrieved through the MolNetEnhancer⁵⁸ workflow and based on similarities in MS2 fragmentation patterns. Relative metabolite abundances are scaled for better pattern visualization.

We believe that chemical differences across different putative classes are now clearly visualized, and we further clarified the sentence in the main manuscript on page 7 highlighted by the reviewer to:

“A number of compound classes that may be linked to compounds of bacterial origin - such as acylcarnitines (organonitrogen compounds), glycerophospholipids and bile acids (steroids and steroid derivatives) have been found to have different abundances in the two armpits (Supplementary Figure 4).”

In addition, we added a detailed description to the material and methods section, where we describe how the heatmap was created as well as how rows and columns were clustered (euclidean distance and complete clustering method):

A Kruskal-Wallis test was used to find differentially abundant metabolites across left and right armpits and *P* values were adjusted for multiple hypothesis testing using the false discovery rate (FDR) method [ref]. To visualise different distributions of metabolites across left and right armpits a heatmap of the differentially abundant metabolites (FDR-adjusted *P* < 0.05) was created using the ComplexHeatmap package version 2.8.0 [ref] in R. Rows and columns were clustered using the euclidean distance and complete clustering method. Only differentially abundant metabolites with a putative class annotation were displayed in the heatmap. The Jupyter notebook used to create the heatmap can be found at <https://github.com/>.

To clarify how putative metabolite annotations were retrieved, we have furthermore added an additional paragraph in the Methods section called “Metabolite annotation”:

Metabolite annotation

MZmine preprocessed MS/MS fragmentation spectra were submitted to feature-based mass spectral molecular networking through the Global Natural Products Social Molecular Networking Platform (GNPS) [refs] and searched against all GNPS spectral libraries. To further enhance chemical structural information we performed *in silico* structure annotation using Network Annotation Propagation [ref] and created consensus chemical classes per molecular family using the GNPS MolNetEnhancer workflow (<https://ccms-ucsd.github.io/GNPSDocumentation/molnetenhancer/>) [ref] with chemical class annotations retrieved from the ClassyFire chemical ontology [ref]. The GNPS molecular networking job can be accessed at <https://gnps.ucsd.edu/ProteoSAFe/status.jsp?task=dd4ed24be55b4e0c96d542a12b7b464e;> *in silico* Network Annotation Propagation results at <https://proteomics2.ucsd.edu/ProteoSAFe/status.jsp?task=4cd9cd5b7f94463abc8edd4aca94c415> and MolNetEnhancer results at <https://gnps.ucsd.edu/ProteoSAFe/status.jsp?task=d341e23d0aa8489d9776118c48db5a67>.

- Supplementary Figure 5A: add “(qPCR)” to the axis title: “16S copies per uL (qPCR)”.

A: Done, thank you

- Supplementary Figure 5B: write in the caption what C, UB, LB, CLA, CRA stands for.

A: Caption now reads:

b) Bacterial load in all samples separated by sampled volunteers and displayed as a ratio of non-silver to silver. All samples falling below the dotted line indicate the increase of the bacterial load during the silver phase. Abbreviations: C=Chest, UB=Upper Back, LB=Lower Back, CLA=Clothes Left Armpit, CRA=Clothes Right Armpit.

- Supplementary Figure 6: in the legend for letter D *Cutibacterium* is not in italic. There is a typo in letter F: “relativa”.

A: Thank you for catching these. Adjusted.

- Supplementary Figures 6D, 8B and 8D: I strongly suggest that authors change the red/green combination to a color combination that is colorblind-friendly.

A: We have adjusted colors on Supplementary Figure 6D and 8D(Now 8B). We have also removed Supplementary Figure panel 8A and B since they were redundant with the information presented in Supplementary Figure 7

- Supplementary Figure 8B: what does “personal” mean?

A: We have removed Supplementary Figure 8 panels A and B since they convey redundant information presented in Supplementary Figure 7

- Supplementary Figure 9: standardize the display of p-values to either scientific format or decimal form.

A: This is now adjusted, thank you

- Supplementary Figure 10D: what is the time point of this comparison between armpit side?

Reviewer #2 (Comments for the Author):

This is a very interesting work that brings some information about the composition of the skin microbiota under the effect of fabrics containing silver with antimicrobial claims. The experimental design was well executed and the results were quite promising, but I had some questions about the discussions.

Regarding chemical diversity, it was not clear to me how this result was obtained, I understood that the Shannon diversity index was used, but exactly how was the index calculated? For example, looking at figure 1 a) I would expect a greater number of only green nodes, showing that there is a greater number of molecules without the presence of silver. But what I see seems to be the opposite, the difference is small, but a greater amount of only grey nodes are observed, which would indicate a greater production of different molecules in the presence of silver, correct?

The other observations I found all very interesting and consistent with the data. Just out of curiosity, did you investigate other influences such as climate, which could be equally affecting all the volunteers in the different periods (with silver and without silver clothes). Because as the two periods (groups) were run at different times, maybe the results could be masked by a variable that affects everyone equally.

A: We have added this crucial information in the Study Design section of M&M:

“Participants all lived in the temperate climate and region of San Diego during the time of study (April-June 2017).”

Did you collect sensory information from the participants? If they noticed a difference in odours, skin appearance, etc.

A: Very good question - we have not performed odour assessments on the shirts or participants, as it was not part of the study design. We have noted down remarks and feedback participants provided, such as for example acne formation or skin feel. These provided us with meaningful insights.

Did you do any evaluation of volatile compounds?

A: We have performed LC/MS, accounting for mainly the non-volatile compounds. Volatile compounds were also captured with the sampling, but were incomplete and unreliable at this point to draw meaningful conclusions. In a follow-up study, this would definitely be a meaningful parameter to take into account.

December 16, 2022

Dr. Pieter C. Dorrestein
University of California, San Diego
Department of Pharmacology, Chemistry and Biochemistry
Skaggs School of Pharmacy and Pharmaceutical Sciences
La Jolla, CA 92093

Re: mSystems00922-22R1 (The molecular effect of wearing silver-threaded clothing on the human skin)

Dear Dr. Pieter C. Dorrestein:

Your manuscript has been accepted, and I am forwarding it to the ASM Journals Department for publication. For your reference, ASM Journals' address is given below. Before it can be scheduled for publication, your manuscript will be checked by the mSystems production staff to make sure that all elements meet the technical requirements for publication. They will contact you if anything needs to be revised before copyediting and production can begin. Otherwise, you will be notified when your proofs are ready to be viewed.

Publication Fees:

If you would like to submit a potential Featured Image, please email a file and a short legend to msystems@asmusa.org. Please note that we can only consider images that (i) the authors created or own and (ii) have not been previously published. By submitting, you agree that the image can be used under the same terms as the published article. File requirements: square dimensions (4" x 4"), 300 dpi resolution, RGB colorspace, TIF file format.

We recognize that the video files can become quite large, and so to avoid quality loss ASM suggests sending the video file via <https://www.wetransfer.com/>. When you have a final version of the video and the still ready to share, please send it to mSystems staff at msystems@asmusa.org.

Congrats to all team!

Sincerely,

Carla Porto
Editor, mSystems

Journals Department
Fig.S10: Accept
Supplemental Material: Accept
Supplemental Material: Accept
Supplemental Material: Accept
Fig.S8: Accept
Supplemental Material: Accept
Supplemental Material: Accept
Supplemental Material: Accept
Fig.S9: Accept
Fig.S1: Accept